# THE RELIABILITY GAP IN AGENTIC EVIDENCE VERIFICATION FOR MATERIALS SCIENCE

Albert Gong[1,*] James J. Kim[1], Anmol Kabra[1], Aaditya Panigrahi[1], Jiashuo Wang[2],
Arjun B. Mulchandani[1], Michael Freeman[1], Fatmagül Katmer[3], Joshua P. Wakefield[3],
Linxi Zhao[1], Chao Wan[1], Akanksha Sarkar[1], Yoav Artzi[1], Leslie M. Schoop[3],
John Thickstun[1], Kilian Q. Weinberger[1], Eun-Ah Kim[1], Peter Frazier[1], Jennifer Sun[1]

[1]Cornell University     [2]Boston University     [3]Princeton University

## ABSTRACT

The explosive growth of scientific literature has created an urgent need for autonomous agents capable of rigorous evidence verification. However, existing evaluation frameworks largely rely on sanitized oracle settings (e.g. pre-parsed PDFs) or controlled knowledge environments (e.g. curated corpus), which abstracts away the noise and complexity of real-world research. We develop a framework for evaluating agentic evidence verification under realistic conditions, grounding our study on a set of 200 open-source papers from the SuperCon database. We select materials science as our testbed because it demands exact numerical grounding and binary factual certainty—strict constraints that expose model reliability issues. We develop our evaluation criteria on accuracy and reliability with material science experts, focusing on two core workflows: Multi-Property Extraction from raw, multimodal PDFs, and Open-World Precedent Search on the live web, explicitly testing the ability to verify "negative results". Systematic study of general-purpose agents reveals a sobering reality: models often identify relevant properties or materials but fail to reliably extract precise values or ensure that answer correctness is grounded with valid attribution—highlighting a fundamental reliability gap in current systems.

## 1  INTRODUCTION

The rate of scientific production is rapidly outpacing our human capacity to synthesize it. With the volume of literature growing exponentially, validating experimental precedents and verifying negative results has become a prohibitively manual bottleneck across many fields of science (Bornmann & Mutz, 2015; Hanson et al., 2024). While Large Language Models (LLMs) and autonomous agents are urgently needed to scale this process, their deployment is currently compromised by a lack of factual reliability.

Scientific evidence demands a level of rigorous source grounding and exact numerical verification that existing models struggle to guarantee. Consequently, we study the open question:

*To what extent can general-purpose agents meet the standards of evidence verification required by scientists for real-world use?*

While the field has made significant strides in the study of LLMs and agentic frameworks, existing evaluations often abstract away the practical challenges of evidence retrieval. Prior work has largely focused on scientific literature understanding (Asai et al., 2024; Laurent et al., 2024; Skarlinski et al., 2024) and reasoning tasks (Rein et al., 2024; OpenAI, 2025; Zhang et al., 2025). However, these evaluations largely operate in oracle settings where relevant, sanitized evidence is provided in advance, or focus on controlled knowledge rather than dynamic retrieval from the open web.

Instead, we ground our study on a set of 200 papers that have been meticulously curated by domain experts, as part of the SuperCon database Materials Database Group (2022). We focus on real-

---

*Corresponding author: `agong@cs.cornell.edu`

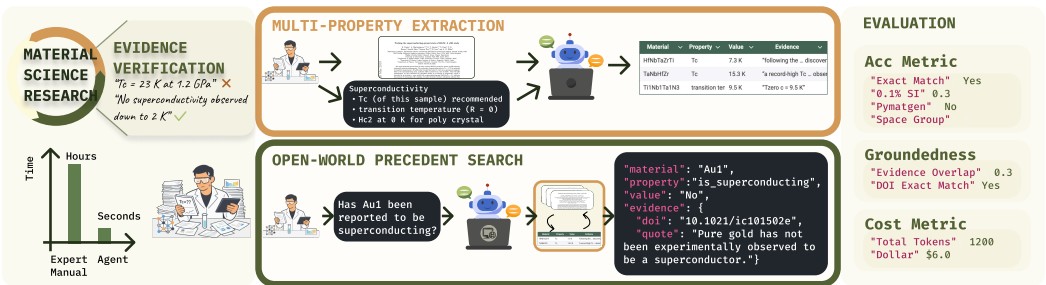

Figure 1: **Overview**. Validating experimental evidence in materials science often requires hours of expert effort, whereas agents can perform similar verification in seconds. Our tasks, multi-property extraction and open-world precedent search, evaluate the reliability gap of agentic workflows in materials science.

world evidence verification tasks that represent fundamental bottlenecks in scientific research. We propose materials science as the ideal diagnostic testbed to stress-test these capabilities. Unlike fields where answers can be subjective, materials science is grounded in physical reality: a material either superconducts given experimental conditions or it doesn't, and properties like critical temperature require exact numerical values with specific units. In this domain, a single hallucinated precedent can lead to months of wasted laboratory time. Our tasks are:

1. **Multi-Property Extraction**: The agent must act as a precise data curator. Starting with a full-text scientific PDF (containing text, tables, and figures), it must extract specific scalar properties (e.g., Critical Temperature $T_c$) along with their correct units. This requires more than simple keyword matching; it demands resolving synonymous entities, filtering out theoretical predictions in favor of experimental facts, and handling complex document layouts.

2. **Open-World Precedent Search**: The agent must act as a rigorous literature reviewer. It is tasked with verifying whether a candidate material has been previously tested for a specific property or synthesis method by searching the open internet and scientific publication databases to retrieve and analyze relevant literature. This is significantly harder than standard retrieval because it requires reliable *abstention*: the ability to correctly confirm a "negative result"—that no evidence exists in the open literature—and to distinguish between absence of evidence and evidence of absence.

These are routine, daily necessities for every lab, yet they remain tedious and time-consuming for human scientists. As they require only "reading" and "seeking", they should be ideally suited for LLMs. However, based on an extensive evaluation of existing models, we observe a sobering reality: current general-purpose agents are not yet ready for the lab. Despite their fluency, they struggle with "evidence-grade" verification in open-world settings. We find that these models frequently fail to retrieve long-tail information, hallucinate citations to sound plausible, and achieve consistently abysmal recall rates. Challenges such as resolving name variations (e.g., chemical synonyms), distinguishing confirmed experimental facts from background discussion, and reliably abstaining when no evidence exists prove surprisingly difficult for current models. Crucially, we observe that increasing agentic complexity (e.g., sophisticated loops and planning) often provides negligible gains over simpler workflows, suggesting that the core reliability gap lies in fundamental retrieval and grounding capabilities rather than insufficient "reasoning" depth.

In this paper, we present the following contributions:

- We design and operationalize a comprehensive evaluation framework constructed end-to-end with materials scientists, based on papers from SuperCon. We work closely with domain experts to ensure our evaluation tasks mirror the noisy, multi-modal reality of scientific literature. We release this as a challenging, high-value resource for the machine learning community, offering a testbed rooted in real-world scientific workflows.
- We identify and define a rigorous scoring rubric that goes beyond standard NLP metrics. Our metrics for multi-property extraction incorporate high accuracy requirements specified

by material scientists; for open-world precedent search, we evaluate both accuracy and grounding, and explicitly assess the reliability of "negative" assertions.
- We provide a systematic study of general-purpose agentic paradigms, from zero-shot prompting to Terminus-2, demonstrating that improved agentic scaffolding does not reliably translate into grounded attribution or faithful scientific extraction.

## 2 RELATED WORK

### 2.1 EVALUATING LLMs FOR SCIENTIFIC RESEARCH

**Evaluating Scientific Knowledge and Reasoning** A large body of prior work evaluates LLMs on scientific understanding and reasoning, including question answering over curated documents, summarization, and factual extraction (Asai et al., 2024; Skarlinski et al., 2024). As model capabilities have advanced, these evaluations have expanded to long-form responses and knowledge-intensive scientific questions (Asai et al., 2024; Han et al., 2024; Upadhyay et al., 2025; Zhao et al.; Rubungo et al., 2025), including PhD-level problem solving (Rein et al., 2024).

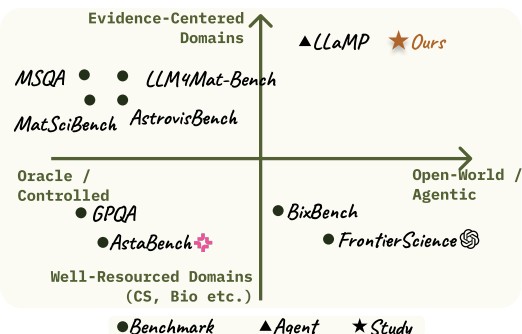

Figure 2: Positioning of our benchmark relative to prior work along two axes: evaluation setting and scientific domain. While prior work focuses on well-resourced domains and oracle-style evidence, we study the reliability gap in materials science.

Related benchmarks assess scientific domain knowledge and reasoning across disciplines, such as biology and medicine (Laurent et al., 2024; Singhal et al., 2023). For example, GPQA (Rein et al., 2024) evaluates graduate-level multiple-choice questions, Frontier Science (OpenAI, 2025) targets expert-level reasoning across physics, chemistry, and biology, and MatSciBench (Zhang et al., 2025) focuses on materials science. While informative, these benchmarks largely operate in oracle settings, where relevant evidence is curated and provided in advance.

**Evaluating Scientific Agentic Workflows.** Real-world scientific research involves iterative information seeking and evidence verification, motivating growing interest in LLM-based agents (Yao et al., 2022; Schick et al., 2023; Mialon et al., 2023). Recent benchmarks evaluate such agentic workflows in well-resourced domains, including computer science (Bragg et al., 2025; Kon et al., 2025) and computational biology (Mitchener et al., 2025; Xuefei et al., 2025). For example, AstaBench (Bragg et al., 2025) assesses agentic abilities such as paper discovery and citation analysis in controlled literature settings.

However, as illustrated in Figure 2, comparable benchmarks remain limited in materials science. While controlled "exam-style" evaluations provide clean signals of isolated capabilities, they do not capture open-world evidence discovery and verification. In contrast to task-specific scientific assistants (Singh et al., 2025), we evaluate general-purpose LLM agents in materials science workflows, focusing on whether broadly available agentic frameworks can meet reliability requirements without domain-specific customization.

### 2.2 LLMs FOR MATERIALS SCIENCE RESEARCH

LLMs are increasingly used in materials science to support research workflows, particularly for extracting structured material properties from the scientific literature (Panapitiya et al., 2021; Gupta et al., 2024; Ghosh & Tewari, 2025; Itani et al., 2025; Dagdelen et al., 2024). While task-specific extraction pipelines can perform well in narrow settings (Yan et al., 2022; Shetty et al., 2023; Polak & Morgan, 2024), they often require substantial manual engineering and do not generalize across properties or domains. We therefore focus on general-purpose LLM agents.

Beyond extraction, materials research frequently requires precedent search, identifying whether prior studies report specific materials or properties. These workflows remain largely manual and are especially challenging in superconductors, where key properties such as critical temperature are numerically precise, condition-dependent, and sparsely reported.

Motivated by these use cases, we study two core materials science workflows: multi-property extraction and precedent search. Using the SuperCon database, we evaluate whether general-purpose LLM agents can reliably extract and identify the evidence needed to support scientific claims in practice.

## 3 TASK SETUP

To systematically evaluate LLMs and general-purpose autonomous agents, we operationalize two complementary materials science workflows in the superconductivity domain: *Multi-Property Extraction* and *Open-World Precedent Search*.

**Dataset.** SuperCon is the largest publicly available database of superconducting materials, built and maintained by the National Institute for Materials Science (NIMS) in Japan through extensive manual curation Materials Database Group (2022). It aggregates roughly 33,000 experimentally studied superconductors drawn from nearly 7,500 publications and is widely used as a de-facto benchmark in data-driven superconductivity research. For each material, SuperCon records composition, basic metadata, and a broad range of superconducting properties, comprising on the order of 100 possible fields with an average of about nine reported values per entry, including but not limited to the critical temperature ($T_c$). These properties are extracted from a combination of main texts, figures, and supplementary materials, resulting in a large and heterogeneous dataset spanning compositions, metadata, and a wide range of experimentally reported superconducting properties.

From this, we query the Crossref API[1] to retrieve DOI information for ∼6,300 publications using their associated metadata in the SuperCon corpus. By cross-referencing these DOIs, we identify and procure 1,334 publicly available papers from arXiv. From this pool, we randomly sample 200 papers to construct our evaluation set. Given that SuperCon consists of historical literature with diverse formatting and multimodal data extracted from figures, text, and supplementary materials, it provides a rich testbed for multi-property extraction.

### 3.1 TASK I: MULTI-PROPERTY EXTRACTION

**Task Setup.** We formulate property extraction as a long-context, multi-modal reading comprehension task. Given a full-text scientific PDF containing text, figures, and tables, the agent must output a structured list of properties that satisfy a specific schema. Unlike standard reasoning benchmarks, this task strictly evaluates instruction following and fidelity, where agents must retrieve exact values without hallucinating or inferring unreported data. As such, this task evaluates an agent's ability to reconstruct scientific evidence from primary sources under strict fidelity constraints, rather than merely populating predefined fields.

The agent operates over papers referenced in SuperCon and extracts properties from a closed set of predefined categories, spanning structural, mechanical, thermal, electrical, and magnetic categories, producing schema-conformant outputs directly from the source documents.

**Evaluation.** Existing metrics for short-form question-answering tasks are not well-suited for the property extraction task. Given that the agent must extract all targeted properties , the number of properties varies across papers. While SuperCon is the most comprehensive database available, it may not record every property present in the source PDF; thus, our recall scores represent the agent's ability to match human-curated highlights. Each property contains a name string and a value string, which includes the numerical value and unit (when applicable). The scoring rule for each property value must depend on the property name.

We propose a two-stage procedure to address these challenges. Given two sets of properties: (1) We embed each property name into a vector and use cosine similarity to find the top-$K$ candidates from

---

[1] https://www.crossref.org/documentation/retrieve-metadata/rest-api/

the opposite set (2) We then use an LLM as judge to find the best match. (See Appendix C for full details on the embedding model and matching prompt).

Within each pair of predicted and ground-truth properties, we use the ground-truth to determine the appropriate scoring rule to apply. To account for the high accuracy requirements in this domain, we use a 0.1% SI unit threshold for numerical values (as determined by domain experts in superconductivity), exact match for categorical values, and near match after normalization for chemical formulas using the `pymatgen` library.

## 3.2 Task II: Open-World Precedent Search

**Task Setup.** Given a material name, the agent must determine whether superconductivity has been reported or abstain with an explicit `Unknown` label. For positive cases, it extracts all reported critical temperature $T_c$ values (e.g., 5 K); for negative cases, it reports the lowest tested temperatures $T_{cn}$ at which superconductivity was experimentally ruled out. All claims must be supported by source DOIs, and agents additionally retrieve auxiliary metadata (e.g., electronic phases, related materials) to provide grounding signals for failure diagnosis. As such, performance primarily reflects technical reading comprehension, evidence retrieval, and adherence to task instructions, rather than multi-step abstract reasoning, evaluating an agent's ability to retrieve, ground, and faithfully extract factual scientific information from external sources.

Because web-based scientific search tasks are costly and reproducible benchmarking requires verifiable attribution, we construct a $T_c$-`dev-set` of 200 materials from SuperCon with publicly accessible DOI references, balanced between 100 superconducting and 100 non-superconducting instances.

**Evaluation.** Model outputs are evaluated by comparing extracted values against ground-truth annotations. For each material, a prediction is considered correct if any reported value ($T_c$ or $T_{cn}$) matches the ground truth. Although all evaluation materials are labeled as superconducting or non-superconducting, agents may abstain by predicting `Unknown` when evidence is insufficient.

Predictions are scored in $\{0, 1, 2\}$. A score of 2 requires both correct binary classification of whether the material is superconducting and the correct reported value ($T_c$ for superconductors, $T_{cn}$ for non-superconductors). A score of 1 is assigned for correct classification alone. A score of 0 is assigned for incorrect classification (including `Unknown`), task failures, or malformed outputs that violate the JSON schema defined in F. For diagnostic analysis, we refer to a score of 2 as *Correct*, 1 as *Partial*, and further decompose score-0 outcomes into *Not Attempted* (valid abstention via `Unknown`), *Incorrect* (wrong classification), and *Failed* (execution or schema errors).

To ensure comparability across agent harnesses, prevent unbounded search, and control API costs, we impose a 20-minute wall-clock limit per trial. This threshold was informed by consultations with three PhD-level scientists, each of whom completed the task within 20 minutes using web-search tools. In practice, most trials finished well below this limit (Table 8).

## 4 Agent Setup

We evaluate general-purpose large language models and agentic systems on our two evidence verification workflows under controlled yet realistic conditions. Our goal is to measure practical reliability in open-world scientific settings, while ensuring fair comparisons across models with heterogeneous tool access.

### 4.1 Models

On the multi-property extraction task, we focus our evaluation on two frontier, multi-modal models: `gemini-3-pro` and `gpt-5.2`. For a performance comparison across model sizes, we additionally include `gemini-3-flash` and `gpt-5-mini` in Appendix A.1. On the open-world precedent search task, we use `gpt-5.1` instead of `gpt-5.2` and further evaluate the text-only `qwen3-max`. All models are treated as black-box APIs with no task-specific finetuning.

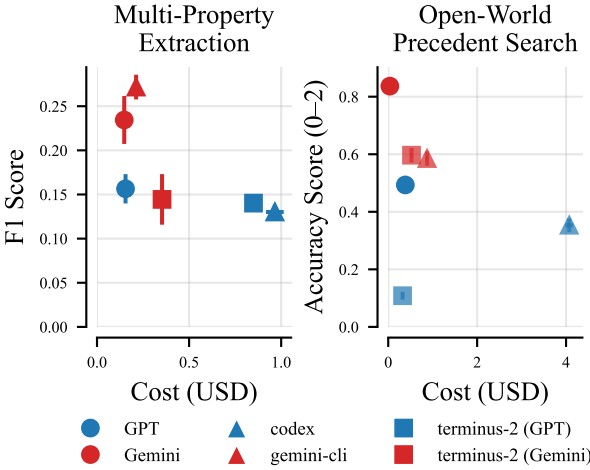

Figure 3: **Accuracy versus cost.** Costs are estimated from OpenAI and Gemini API pricing, except for `T2`. For multi-property extraction, we report mean ± 1 standard error F1 score for 200 tasks across one run. For open-world precedent search, we report the mean ± 1 standard error accuracy score for 200 tasks across three runs. See Figure 10 for accuracy versus total tokens. See Section 4 for model and agent harness details.

## 4.2 AGENT FRAMEWORKS

We conduct all experiments using Harbor, a sandboxed evaluation framework for agentic systems that provides isolated Linux environments and trajectory logging (Merrill et al., 2026). This allows us to track each agent's actions (e.g., tool calls, web queries, execution code) and directly attribute performance to specific behaviors. We evaluate two classes of agents, under the reasoning settings detailed in Section E. For model versions, see Appendix E

**Integrated Harness Agents.** These agents pair models with their native tooling ecosystems. Gemini models are evaluated using the `gemini-cli` harness with access to Google Search; GPT models use the `codex` agent; and Qwen models use `qwen-code`. These configurations reflect how each system is typically deployed in practice, combining LLM reasoning with proprietary harnesses and tools.

**Terminus Agents.** To enable controlled comparisons across models, we additionally evaluate Gemini and GPT models within Terminus-2 (`T2`), Harbor's reference agent implementation (Merrill et al., 2026). `T2` restricts agents to terminal-based interaction inside a Linux shell, without access to proprietary search APIs or privileged tools. In this setting, agents must rely solely on standard command-line utilities (e.g., `curl`, `wget`) for retrieval.

This dual evaluation protocol allows us to separately measure (i) end-to-end performance in realistic, tool-augmented deployments and (ii) core retrieval and grounding capability under a uniform execution environment.

## 5 RESULTS

### 5.1 WHAT IS THE MOST COST EFFECTIVE WAY OF DOING EVIDENCE VERIFICATION?

We compare direct prompting against integrated agent harnesses (`gemini-cli`, `codex`, `T2`). `gemini-3-pro` with direct prompting achieves the best accuracy-cost tradeoff for property extraction, while `gemini-3-pro` with `gemini-cli` performs best for precedent search. Notably, `gpt-5.2` with direct prompting achieves the lowest failure rate on both tasks (Figures 4 and 5). Agentic scaffolding generally degrades performance: in most configurations, adding an agent harness reduces accuracy while increasing cost. Two exceptions are `gemini-3-pro` with `gemini-cli`, which improves precedent search accuracy at higher cost, and `gpt-5.1` with `T2`, which reduces cost but substantially hurts accuracy.

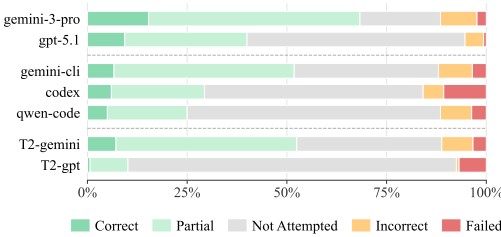
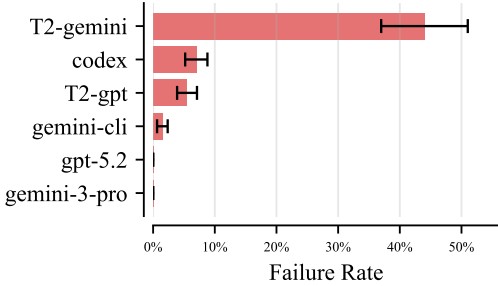

Figure 4: **Precedent search score distribution.** Percentage distribution of scores for 200 tasks across three runs. *Correct*: correct classification and $T_c$/ $T_{cn}$ value; *Partial*: correct classification but wrong value; *Not Attempted*: model output "Unknown"; *Incorrect*: wrong classification; *Failed*: execution error.

Figure 5: **Reliability of agents on multi-property extraction**. We report mean ±1 standard error across 200 tasks and one run. Failure here means that no properties were extracted due to execution errors, including tool call failures, timeout, and output parsing errors.

## 5.2 WHY DO AGENTS STRUGGLE WITH PROPERTY EXTRACTION?

To understand why agents achieve relatively low F1 scores on property extraction, we decompose performance into two components. We define the *property-material match* as the F1 score computed by checking whether each predicted property-material pair has a corresponding match in the ground truth (and vice versa), without requiring the extracted *value* to be correct. This metric isolates the agent's ability to identify *what* to extract from its ability to extract the correct *value*.

Table 1: **Property-material match vs. final F1 score.** Agents correctly identify property-material pairs at a much higher rate than they extract correct values, indicating that value extraction is the primary bottleneck. Mean $\pm$ 1 SE.

| Agent | Model | F1 ↑ | PM Match ↑ |
|---|---|---|---|
| zeroshot | `gemini-3-pro` | $0.27 \pm 0.01$ | $0.43 \pm 0.02$ |
| `gemini-cli` | `gemini-3-pro` | $0.27 \pm 0.01$ | $0.43 \pm 0.02$ |
| T2 | `gemini-3-pro` | $0.14 \pm 0.03$ | $0.22 \pm 0.04$ |
| zeroshot | `gpt-5.2` | $0.18 \pm 0.01$ | $0.35 \pm 0.01$ |
| `codex` | `gpt-5.2` | $0.13 \pm 0.01$ | $0.26 \pm 0.01$ |
| T2 | `gpt-5.2` | $0.14 \pm 0.01$ | $0.26 \pm 0.01$ |

Table 1 reveals a consistent pattern: agents achieve substantially higher property-material match scores than final F1 scores. For example, `gemini-3-pro` with zeroshot prompting achieves a property-material match of $0.43$ but an F1 of only $0.27$—a gap of 16 percentage points. This indicates that agents are reasonably good at identifying *which* properties and materials to extract, but struggle to extract the correct *values*. We investigate in Appendix A the effects of model size, prompting strategies, and model reasoning level on task performance.

**Impact of Ground Truth Granularity.** We observe that model performance is sensitive to the numerical tolerance threshold. At a strict expert-defined threshold of 0.1%, F1 scores remain depressed across all models. However, relaxing the tolerance to 10% yields modest improvements of 2–4 percentage points in F1 score (see Table 6). Qualitative analysis reveals multiple sources of error: (1) agents sometimes extract approximate values from text (e.g., "~6 K" vs. ground truth 6.24 K), (2) OCR artifacts and malformed table layouts lead to incorrect value parsing, and (3) when multiple related values appear in context (e.g., two estimates from different models), agents may select the wrong one. These findings suggest that the gap between agent extractions and ground truth reflects both genuine extraction errors and noise introduced by document processing artifacts.

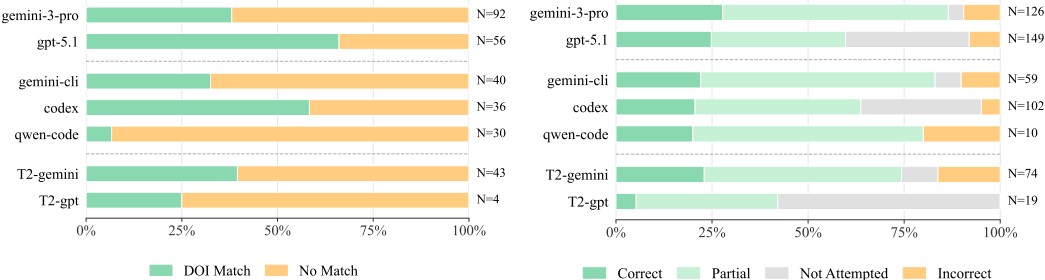

Figure 6: **DOI retrieval rate conditioned on correctness.** Percentage of trials with a DOI match, restricted to cases where the model was correct (class + value). $N$ denotes the number of correct trials, across three runs.

Figure 7: **Score distribution conditioned on successful retrieval.** Percentage distribution of scores over trials where the correct DOI was retrieved, across three runs. $N$ denotes the number of trials where the DOI was correctly retrieved.

## 5.3 CAN AGENTS GROUND PRECEDENT SEARCH IN EVIDENCE?

Correctly identifying whether a material is superconducting and reporting the associated $T_c$ or $T_{cn}$ is necessary but insufficient; scientific use additionally requires explicit grounding in primary sources. Accordingly, we evaluate not only answer correctness, but whether agents reliably *attribute* predictions to the appropriate literature and *extract* accurate values from retrieved sources. To disentangle these capabilities, we study two complementary conditional questions: (1) When an agent produces a fully correct prediction (class + value), does it also cite the correct DOI? and (2) When an agent retrieves the correct DOI, does it accurately extract the relevant scientific evidence? These conditionals expose distinct failure modes in the composite precedent search pipeline, which combines open-world retrieval, technical reading comprehension, and structured property extraction. (See Appendix D for a discussion on ground-truth DOI uniqueness).

**Correct answers are often weakly grounded.** As shown in Figure 6, attribution is unreliable even when predictions are correct. For example, while `codex` grounds a majority of its answers, `gemini-cli` frequently reaches correct conclusions without linking to primary sources—rendering the answers unverifiable. Notably, single-turn models outperform agentic workflows, suggesting that multi-step scaffolding does not consistently improve attribution.

**Successful retrieval does not imply faithful extraction.** Retrieving the correct paper often results in only partial correctness (Figure 7). Agents frequently identify the class correctly but fail to extract the precise $T_c$ or $T_{cn}$, where we again observe limited gains from agentic scaffolding.

**Implications for agentic evidence verification.** These findings highlight a fundamental reliability gap in current general-purpose agents: correctness, attribution, and faithful extraction do not co-occur consistently. Agents may answer correctly without proper grounding, retrieve sources without accurate interpretation, or fail to abstain when evidence is insufficient. Such behavior limits practical utility for precedent verification, motivating the need for evaluations that disentangle retrieval from extraction and for agent designs that more tightly integrate source attribution with accurate scientific interpretation.

## 5.4 ANALYZING AGENT TRAJECTORIES IN PRECEDENT SEARCH

Agents comprise multiple interacting components: LLMs plan multi-step procedures, invoke tools, and synthesize final answers, while search engines return ranked results for valid queries. We analyze agent trajectories on the precedent search task, examining how tool usage patterns vary with prediction correctness.

**How efficiently do agents use tools?** In Table 2 we observe that `gemini-cli` requires the fewest tool calls. This is likely because its integrated Google Search is tightly coupled with `gemini-3-pro` within Google's ecosystem. Crucially, this efficiency does not sacrifice correctness: `gemini-cli` also achieves the highest accuracy (Figure 4). In contrast, `codex` issues the most tool calls despite

sharing the same terminal-based tooling as T2 agents—yet this additional effort does not translate to better *Correct* performance than `gemini-cli` in Figure 4.

| Agent | # Tool Calls |
|---|---|
| `gemini-cli` | $25.1 \pm 0.5$ |
| `codex` | $59.1 \pm 0.5$ |
| `T2-gemini` | $42.4 \pm 1.4$ |
| `T2-gpt` | $38.9 \pm 1.1$ |

Table 2: **Tool calls per agent on the precedent search task.** `gemini-cli` is the most efficient, due to its tightly integrated Google Search tool with the `gemini-3-pro` model. We omit `qwen-code` as it does not log complete tool information. Results aggregated across 3 runs.

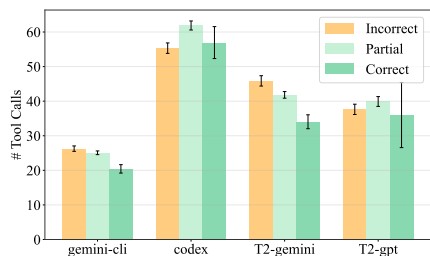

Figure 8: **Tool calls stratified by prediction correctness.** `gemini-3-pro` model uses *fewer* tool calls when it answers correctly (in both `gemini-cli` and `T2-gemini` harnesses), suggesting that successful early search hits allow the model to terminate sooner. Results aggregated across 3 runs.

Figure 8 reveals a counterintuitive insight: `gemini-3-pro` uses fewer tool calls on correct predictions than on incorrect ones. Manual inspection of `gemini-cli` and `T2-gemini` trajectories confirms that `gemini-3-pro` often retrieves the relevant results with Google Search hits early, enabling the agent to terminate without further exploration. `gpt-5.1` exhibits the opposite behavior: it rarely exits early, instead continuing to validate until confident in the citation. This aligns with Figure 6, where `codex` (powered by `gpt-5.1`) achieved the highest DOI match rate among all agents.

**Are all tools equally useful?** Beyond the *quantity* of tool calls, we examine whether *search engine choice* affects agent performance, as APIs differ in result quality and relevance. Fixing the harness to T2, Figure 9 shows search engine usage (fraction of tool calls) stratified by prediction correctness. Although the restricted terminal environment forces both `gemini-3-pro` and `gpt-5.1` to query multiple engines, specific sources consistently dominate among correct trajectories. For `gemini-3-pro` (a), successful runs rely heavily on the `arxiv` API and `duckduckgo`, likely reflecting the absence of Google Search. In contrast, `gpt-5.1` (b) favors `jina.ai` over traditional academic APIs such as `arxiv`. These results suggest distinct model-specific tool preferences, with higher-quality academic sources correlating with improved performance.

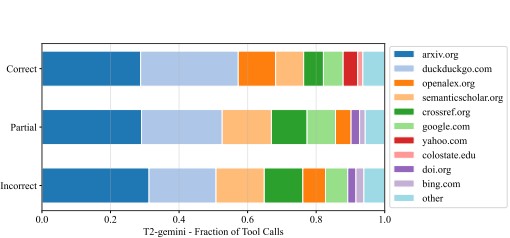

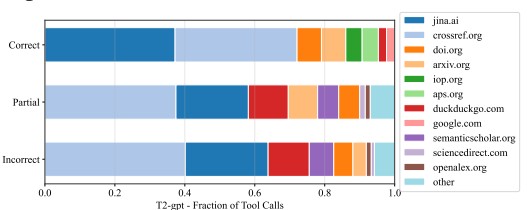

(a) `T2-gemini` agent relies heavily on `arxiv` and `duckduckgo`; their combined share increases with task performance.

(b) `T2-gpt` agent primarily uses `jina.ai` and `crossref`. It references `doi.org` to validate final outputs. Curiously, the agent abandons `semanticscholar` entirely in correct predictions: the model hits API rate limits and fails to retrieve results.

Figure 9: **Models prefer different search engines, and some engines yield better results.** In the T2 harness, models query search engine APIs exclusively through terminal-based commands (e.g., `curl`). Across all agent trajectories in the precedent task, we plot the distribution of search engine calls, and stratify by prediction correctness. Runs are aggregated across 3 runs.

# 6 CONCLUSION

We presented a rigorous evaluation framework for agentic evidence verification in materials science, comprising 200 multi-property extraction tasks and 200 open-world precedent search tasks grounded

in expert-curated papers from the SuperCon database. Our two tasks expose a fundamental reliability gap: current general-purpose agents often identify relevant properties or retrieve candidate sources, yet fail to reliably extract precise values or align correct predictions with grounded attribution. We release our benchmark as a challenging testbed for developing the next generation of reliable, evidence-grounded scientific AI.

## ACKNOWLEDGEMENTS

This work is supported by the National Science Foundation through the AI Research Institutes program Award No. DMR-2433348. This work was partially supported by funding from NewYork-Presbyterian for the NYP-Cornell Cardiovascular AI Collaboration. We thank Modal for sandbox credits and the Gemini Academic Program Award for Google Cloud Credits.

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

# A  ABLATION STUDIES FOR PROPERTY EXTRACTION

## A.1  EFFECT OF MODEL SIZE

A natural question is whether smaller, more resource-efficient models can achieve comparable performance to larger frontier models. This has practical implications for deployment costs and accessibility. We compare flagship models (`gemini-3-pro` and `gpt-5.2`) against their smaller counterparts (`gemini-3-flash` and `gpt-5-mini`) on the property extraction task (Table 3). We use the default thinking level for the Gemini comparison and the same reasoning effort setting for the GPT comparison ("medium").

Table 3: **Effect of model size on property extraction.** F1 score, cost per task (USD), and failure rate. Mean $\pm$ 1 SE across 200 tasks.

| Model | F1 $\uparrow$ | Cost $\downarrow$ | Fail % $\downarrow$ |
|---|---|---|---|
| `gemini-3-pro` | $0.27 \pm 0.01$ | \$0.15 | 0% |
| `gemini-3-flash` | $0.17 \pm 0.01$ | \$0.03 | 32% |
| `gpt-5.2` | $0.18 \pm 0.01$ | \$0.15 | 1% |
| `gpt-5-mini` | $0.18 \pm 0.01$ | \$0.03 | 0% |

For Gemini, `gemini-3-flash` is both less reliable and less accurate than `gemini-3-pro`: the smaller model exhibits a 32% failure rate compared to 0% for the flagship, while also achieving lower F1 (0.17 vs 0.27). In contrast, `gpt-5-mini` achieves the same F1 as `gpt-5.2` (0.18) while being $5\times$ cheaper (\$0.03 vs \$0.15 per task), suggesting that increased model size does not always translate to higher accuracy.

## A.2  PER-PAGE ANALYSIS FOR PROPERTY EXTRACTION

Scientific papers vary significantly in length and complexity. We investigate whether breaking down documents on a per-page basis can simplify the extraction task and improve agent performance. Using `gemini-3-pro`, the best-performing model according to our accuracy-cost analysis (Figure 3), we compare a one-pass workflow that processes the entire document against a page-by-page workflow that extracts properties from each page independently and aggregates results.

The page-by-page workflow extracts substantially more properties on average ($47.55\pm2.19$) compared to the one-pass workflow ($17.20 \pm 0.68$). This increased extraction volume translates to higher recall at $0.36 \pm 0.02$ versus $0.30 \pm 0.02$ for the one-pass method. However, this gain comes at the cost of precision, which drops from $0.27 \pm 0.01$ (one-pass) to $0.19 \pm 0.01$. The resulting F1 scores are 0.27 for one-pass and 0.24 for page-by-page extraction.

## A.3  EFFECT OF REASONING EFFORT

Recent models offer configurable "reasoning effort" settings that control how much compute the model dedicates to chain-of-thought style reasoning before producing a response. We ablate the effect of reasoning effort on `gpt-5.2` for the property extraction task (Table 4).

Table 4: **Effect of reasoning effort on property extraction.** F1 score, cost per task (USD), and failure rate for `gpt-5.2` at different reasoning effort levels. Mean $\pm$ 1 SE across $\sim$200 tasks.

| Reasoning Effort | F1 $\uparrow$ | Cost $\downarrow$ | Fail % $\downarrow$ |
|---|---|---|---|
| Low | $0.20 \pm 0.01$ | \$0.12 | 0% |
| Medium | $0.18 \pm 0.01$ | \$0.15 | 1% |
| High | $0.17 \pm 0.01$ | \$0.19 | 3% |

Contrary to expectations, increasing reasoning effort does not improve F1 on the property extraction task. In fact, performance slightly decreases from 0.20 (low) to 0.17 (high), while cost increases from \$0.12 to \$0.19 per task due to the increased number of completion tokens. Higher reasoning

Table 5: Benchmarks for evaluating LLM-based systems on materials science tasks. *Chatbot* denotes single-turn or conversational LLM usage, while *Agent* denotes multi-step reasoning with tool or database interaction.

| Benchmark | Task | Chatbot | Agent | Size | Input | Output | Evaluation |
|---|---|---|---|---|---|---|---|
| *Property Extraction and Prediction* | | | | | | | |
| ChatExtract (Polak & Morgan, 2024) | Property extraction | ✓ | | 8.9K | Sentence | Extracted values | Precision / Recall |
| Polymer Scholar (Gupta et al., 2024) | Polymer property extraction | ✓ | | ∼1M | Article | Property database | F1, Cost, Time |
| LLaMP (Chiang et al., 2024) | Agentic materials property querying | ✓ | ✓ | ∼1K | Query | Grounded answer | Precision / MAE / Accuracy |
| LLM4Mat-Bench (Rubungo et al., 2025) | Crystalline property prediction | ✓ | | 1.9M | Material description | Property value | MAE / AUC |
| Automated Extraction (Ghosh & Tewari, 2025) | Thermoelectric & structural extraction | ✓ | ✓ | ∼28K | Article | Structured records | Schema & value accuracy |
| *QA* | | | | | | | |
| MaScQA (Bajan & Lambard, 2025) | Undergrad-level materials QA | ✓ | | 0.6K | Question | Answer | Accuracy |
| MSQA (Cheung et al., 2025) | Graduate-level materials QA | ✓ | | ∼1.8K | Question | Answer | Gold answers + LLM judges |
| *Others* | | | | | | | |
| MGB (Yan et al.) | Material generation (crystal, MOF, OOD) | ✓ | | 10K–45K (per task) | Composition / structure | Generated structure | Validity, matching, efficiency |
| MatSciBench (Zhang et al., 2025) | Scientific reasoning | ✓ | ✓ | 1.34K | Question | Solution + answer | Rule-based + LLM judges |
| **Multi-Property Extraction (Ours)** | Full-article structured extraction | ✓ | ✓ | 200 | Article | Structured DB records | F1 (0.1% tolerance) |
| **Open-World Precedent Search (Ours)** | Scientific claim verification | ✓ | ✓ | 200 | Material | Claim + Evidence | Accuracy + DOI attribution |

effort also introduces more failures (3% at high vs 0% at low). We note that the completion length never exceeds the maximum generation length, detailed in Appendix E.

# B  DETAILED RELATED WORK

# C  PROPERTY MATCHING DETAILS

## C.1  EMBEDDING MODEL

For the first stage of property matching, we embed each property name into a 3072-dimensional vector using the `gemini-embedding-001` model with the `SEMANTIC_SIMILARITY` task type. We then compute cosine similarity between predicted and ground-truth property embeddings to identify the top-$K$ candidate matches with $K = 3$.

## C.2  LLM-AS-JUDGE PROMPT

For the second stage, we use the following prompt to determine whether two property descriptions refer to the same physical measurement:

---

### Property Matching Prompt

```
You are an expert condensed matter physicist evaluating whether two property
descriptions refer to the same physical measurement.
## Property 1 - Name:  "{property_name_1}" - Context:  "{context_1}"
## Property 2 - Name:  "{property_name_2}" - Context:  "{context_2}"
## Matching Rules:
SAME property if:
```

- Names are synonymous (e.g., "Tc" = "Critical Temperature" = "Superconducting Transition Temperature")
- Abbreviation differences only (e.g., "Jc" = "Critical Current Density")
- Capitalization/formatting differences

```
DIFFERENT properties if:
```

- Technically distinct measurements:
  - "Tc onset" $\neq$ "Tc zero" $\neq$ "Tc midpoint"
  - "Jc" $\neq$ "Ic" (density vs absolute current)
  - "lattice constant a" $\neq$ "lattice constant c"
  - "upper critical field Hc2" $\neq$ "lower critical field Hc1"
- Different measurement orientations when orientation matters (e.g., "resistivity (c-axis)" $\neq$ "resistivity (ab-plane)")

---

```
        • Different conditions not reconcilable (e.g., "Tc at 0 GPa" ≠ "Tc at
          10 GPa" unless pressure is tracked separately)
## Response Format:  Return JSON only:
{
  "is_match": boolean,
  "confidence": "high" | "medium" | "low",
  "reason": "concise explanation",
  "matched_via": "direct" | "synonym" | "abbreviation"
                 | "condition_reconciliation" | null
}
```

# D   PRECEDENT SEARCH DETAILS

For the precedent search task, the agent is given the prompt in Section F.

**DOI uniqueness.**   A potential concern is whether the ground-truth DOI used for evaluation uniquely identifies the relevant superconducting evidence. In practice, this ambiguity is limited by domain constraints: synthesizing and characterizing candidate superconductors is resource-intensive, requiring substantial experimental effort and specialized infrastructure. As a result, materials reported to be non-superconducting are rarely re-tested independently, since reproducing a negative result typically offers little scientific payoff. Repeat measurements generally occur only when there is strong prior belief in nearby superconducting phases, in which case materials may be revisited as reference systems. Consequently, for the majority of materials in our dataset, the annotated DOI corresponds to the primary experimental report of superconductivity or its absence, making alternative valid sources uncommon.

**DOI and title matching.**   For DOI matching, we perform exact string matching, accounting for both full URL formats (e.g., `https://doi.org/...`) and bare DOI strings. Title matches are determined using a similarity threshold of 0.9 computed via `difflib.SequenceMatcher`.

# E   MODEL SETTINGS

**Model specifications.** The specific LLM model versions we used are as follows: `gemini-3-pro` (`gemini-3-pro-preview`), `gpt-5.2` (`gpt-5.2-2025-12-11`), `qwen3-max` (`qwen3-max-2025-09-23`), `gemini-3-flash` (`gemini-3-flash-preview`), `gpt-5-mini` (`gpt-5-mini-2025-08-07`), `gpt-5.1` (`gpt-5.1-2025-11-13`).

**Reasoning settings.**   We configure each model using its standard reasoning settings for the corresponding workflow. For Gemini 3 Pro, we use the default thinking level ("high").

For the multi-property extraction task, GPT 5.2 is evaluated with reasoning effort set to "medium" in the single-turn workflow. GPT 5.2 is run with the same default ("medium") reasoning configuration when used within the Codex agent harness.

For the open-world precedent search task, GPT 5.1 is evaluated with reasoning effort set to "high" in both the single-turn workflow and within Codex.

All Terminus-2 (T2) agents inherit the reasoning configuration of their underlying LLM as described above. To standardize output capacity across models, we set the maximum generation length to 65,536 tokens for all systems, matching the output limit of Gemini 3 Pro.

## F PROMPTS

---

### SuperCon Property Extraction Prompt

You are a STRICT PHYSICAL PROPERTY EXTRACTION ENGINE.

Your task: Given a scientific paper (text + figures + tables + captions),
extract an EXHAUSTIVE list of physical properties using ONLY information
explicitly stated in the paper. Never infer or guess. If unsure whether
something is a physical property, include it.

Missing information rule: If any field or condition is not stated in the
paper, OMIT the key entirely from the JSON. Do NOT fabricate placeholders.
EXCEPTION: location.page is REQUIRED and must always be a positive integer.

---

TARGET PROPERTIES FOR THIS PAPER

For this paper, you are particularly interested in the following properties.
If you see these properties, make sure to use these standard names and
include as many conditions as possible.

Magnetic property: Neel temperature, Curie temperature, magnetic moment per
formula, temperature independent term in susceptibility, Curie constant

Material: common formula of materials, shape

Mechanical property: density (gcm-3), Young's modulus at 300 K, shear
modulus at 300 K, Poisson ratio at 300 K, sound velocity at 300 K, hardness
at 300 K, shear modulus at 4.2 K, Young's modulus at 4.2 K, Poisson ratio at
4.2 K, sound velocity at 4.2 K

Normal state property: resistivity at RT for poly crystal, resistivity at
normal-T for poly crystal, Hall coefficient at 300 K, resistivity at RT for
single crystal for J//ab-plane, carrier density at 300 K, resistivity at 77 K
for poly crystal, resistivity at normal-T for single crystal for J//ab-plane,
resistivity at 4.2 K for poly crystal, resistivity at 77 K for single crystal
for J//ab-plane, Hall coefficient for single H//c-axis, resistivity at RT for
single crystal for J//c-axis, resistivity at normal-T for single crystal
for J//c-axis, resistivity at 4.2 K for single crystal for J//ab-plane,
resistivity at 77 K for single crystal for J//c-axis, Hall coefficient at 300
K for single H//c-axis, resistivity at 4.2 K for single crystal for J//c-axis,
Hall coefficient at 300 K for single H//ab-plane

Preparation: raw materials, preparation method, target material, substrate

Structure: common name of structure, lattice constant a, lattice constant c,
crystal structure/symmetry, space group, lattice constant b, international
table number, pressure dependence of LATA, pressure dependence of LATC,
temperature dependence of LATA, temperature dependence of LATC, pressure
dependence of LATB, temperature dependence of LATB

Superconductivity: Tc (of this sample) recommended, transition temperature
(mid point), Tc from susceptibility measurement, lowest temperature for
measurement (not superconducting), transition temperature (R = 100%),
transition temperature (R = 0), Hc2 at 0 K for poly crystal, transition width
for resistive transition, -slope in Hc2 vs T at Tc for poly crystal, slope at
P = 0 in Tc vs P plot, volume fraction of Meissner effect(%), normalized
energy gap at 0 K 2$\Delta$(0)/kTc, coherence length at 0 K for poly crystal,
penetration depth at 0 K for poly crystal, energy gap at 0 K $\Delta$(0), Hc2 at
0 K for single crystal for H //c-axis, Hc1 at 0 K for poly crystal, coherence
length at 0 K for single crystal for H //ab-plane, penetration depth at 0
K for single crystal for H //ab-plane, alpha in Tc = A * M^(-alpha) isotope
effect, Hc2 at 0 K for single crystal for H //ab-plane, isotope element,
-slope in Hc2 vs T at Tc for single crystal for H //c-axis, -slope in Hc2 vs T
at Tc for single crystal for H //ab-plane, coherence length at 0 K for single
crystal for H ⊥ab-plane, exchange ratio of isotope(%), DTC = Tc - Tc0 for

isotope element, penetration depth at 0 K for single crystal for H⊥ab-plane, Hc2 at given temperature for poly crystal, Hc1 at given temperature for poly crystal, Hc1 at 0 K for single crystal for H //c-axis, Hc1 at 0 K for single crystal for H //ab-plane, Jc at T = 77 K H = 0 T, Hc2 at given temperature for single crystal H//ab-plane, Jc at 4.2 K H = 0 T, Hc2 at given temperature for single crystal H//c-axis, Hc1 at given temperature for single crystal H//c-axis, Hc1 at given temperature for single crystal H//ab-plane

Thermal property: coefficient of electronic specific heat, Debye temperature, thermopower at 300 K, specific heat jump at Tc ($\Delta$C), thermal conductivity at 300 K, thermopower at 300 K for parallel to ab-plane, thermal conductivity at 300 K for heat flow//ab-plane, thermal conductivity at 300 K for heat flow//c-axis

---

SYMMETRY RULES (VERY IMPORTANT)

---

When the property is about symmetry (space group, point group, crystallographic class, magnetic space group):

1. property_name must be one of: ''space group'', ''point group'', ''crystallographic class'', ''magnetic space group'', etc.

2. value_string must be ONLY the symmetry label (e.g., ''P63mc'', ''Fm-3m'', ''C6v'', ''P4/nmm'').

3. If the text includes description such as ''hexagonal'' or ''noncentrosymmetric'', put that in notes, NOT in value_string.

STRUCTURE PROTOTYPE RULES: If the property is a structural prototype, use: property_name = ''structure prototype'' OR ''crystal structure type''; value_string = full descriptive phrase (e.g., ''Th7Fe3-type hexagonal structure'')

---

LOCATION / GROUNDING (MANDATORY)

---

Every property MUST include:

- location.page (REQUIRED)
- location.section (if available)
- location.figure_or_table (if applicable)
- location.source_type (text, figure, caption, table)
- location.evidence (exact quote --- IT MUST EXACTLY MATCH THE PAPER)

---

OUTPUT FORMAT

---

Return a SINGLE valid JSON payload containing an array of properties:

```json
{
  "properties": [
    {
      "id": "prop_001",
      "material_or_system": "...",
      "property_name": "...",
      "category": "...",
      "value_string": "...",
      "value_unit": "",
      "qualifier": "...",
      "value_detail": "...",
      "conditions": {
        "temperature": "...",
        "pressure": "...",
        "field": "...",
        "frequency": "...",
        "orientation": "...",
        "environment": "...",
```

```
        "sample_state": "...",
        "other_conditions": "..."
      },
      "method": "...",
      "model_or_fit": "...",
      "location": {
        "page": 1,
        "section": "...",
        "figure_or_table": "...",
        "source_type": "text",
        "evidence": "..."
      },
      "notes": "..."
    }
  ]
}
```

---

STOICHIOMETRIC FORMULA RULES

---

The material_or_system field must contain a fully resolved chemical formula with explicit numerical subscripts for every element. No variables, placeholders, or generic notation allowed.

Required format: All elements must be standard chemical symbols (e.g., Y, Ba, Cu, O, La, Sr, Bi, Ca). All subscripts must be explicit integers or decimals (e.g., 2, 0.15, 6.93). Parentheses are allowed for groupings (e.g., (Sr,Ca) becomes explicit values).

Transformations required: $YBa_2Cu_3O_{7-\delta}$ ($\delta$=0.07) $\rightarrow$ YBa2Cu3O6.93; $La_{2-x}Sr_xCuO_4$ (x=0.15) $\rightarrow$ La1.85Sr0.15CuO4; YBCO $\rightarrow$ YBa2Cu3O7 (or O6.93 if specific $\delta$ given); RE-123 (RE=Gd) $\rightarrow$ GdBa2Cu3O7

What to exclude (use generic_formula field in notes if needed): Variables (x, y, $\delta$, n), Ranges (0<x<0.3), Unresolved placeholders (RE, M, Ln), Approximate notation ($\sim$7, 7-$\delta$ without $\delta$ value).

If the paper provides a property for a generic formula without specifying composition, set material_or_system to the most specific formula extractable and document the generic form in notes field.

---

OTHER RULES

---

- Keep units inside value_string (no separate value_number field).
- location.page is mandatory; all other fields are optional---omit if not stated.
- Do not invent values.
- Print only the JSON as your final response.
- Never merge values unless they form a tuple (e.g., coordinates).

FINAL INSTRUCTIONS

1. Scan the entire paper (text, formulas, tables, figures, captions).
2. Extract EVERY explicitly reported target property.
3. Apply symmetry rules, prototype rules, and atomic coordinate rules exactly.
4. Use the output format above (value_string carries any units inline).
5. Do NOT skip or compress entries.
6. Output ONLY the JSON, with no explanations.

## $T_c$ Precedent Search Prompt

You are a careful condensed-matter physicist and materials scientist.

Your task is to determine whether the material **material** has been reported to be superconducting, and to extract detailed properties about its phase behavior and related materials.

You must search for experimental evidence in scientific literature.

Please answer the following questions:

Q1. Has **material** been reported to be superconducting? (Answer "Yes" or "No" or "Unknown")
Q2. If Yes, what are ALL the measured superconducting transition temperatures (Tc) reported? (Provide values designated in Kelvin as a list, e.g., ["12", "11.5"]. Do NOT include unit.)
Q3. If No, what are ALL the lowest temperatures at which it was measured and found NOT to be superconducting (Tcn)? (Provide values designated in Kelvin as a list, e.g., ["0.05", "4.2"].)

**Output Format:**

You must output a SINGLE JSON object exactly matching this structure:

```json
{{
  "properties": [
    {{
      "material": "{material}",
      "property_name": "is_superconducting",
      "value_string": "Yes" | "No" | "Unknown",
      "source_dois": [
        {
          "title": "...",
          "authors": ["..."],
          "year": 2000,
          "doi": "...",
          "quoted_span": "..."
        }
      ]
    }},
    {{
      "material": "{material}",
      "property_name": "tc",
      "value_string": "...",
      "conditions": {{
          "pressure": "...",
          "magnetic_field": "..."
      }},
      "source_dois": [
        {
          "title": "...",
          "authors": ["..."],
          "year": 2000,
          "doi": "...",
          "quoted_span": "..."
        }
      ]
    }},
    {{
      "material": "{material}",
      "property_name": "tcn",
      "value_string": "...",
      "conditions": {{
```

```
            "pressure": "...",
            "magnetic_field": "..."
        }},
        "source_dois": [
          {
            "title": "...",
            "authors": ["..."],
            "year": 2000,
            "doi": "...",
            "quoted_span": "..."
          }
        ]
    }}
  ],
  "superconductors_mentioned": [
    {{
      "material": "...",
      "Tc": "..."
    }}
  ],
  "electronic_or_magnetic_phases": [
    {{
      "material": "...",
      "phase": "...",
      "temperature": "...",
      "temperature_type": "transition | measurement | null"
    }}
  ],
  "missing_or_notable_information": "..."
}}
```

Rules & Constraints:

1. **Core Superconductivity Check (Q1-Q3):**

   - **One Object Per Measurement**: For `tc` and `tcn`, create a **separate** JSON object in the `properties` list for EACH distinct measurement found (with its specific conditions and DOIs). Do not list them.
   - **If Q1="No"**: Do not output any `tc` properties. Provide `tcn` properties for all non-superconducting measurements found.
     - **Tcn Inference**: If the material is reported as not superconducting (e.g., insulating, paramagnetic), provide the lowest temperature measured in the experiment. Do NOT skip just because the text doesn't explicitly say "not superconducting down to X". Infer X from the measurement range.
     • **If Q1="Yes"**: Do not output any `tcn` properties. Provide `tc` properties for all superconducting measurements found.

2. **Superconductor Mention Check (`superconductors_mentioned`):**

   - Identify all materials mentioned that are closely related to **material** AND explicitly stated to be superconductors.
   - Extract material name and Tc (exactly as stated, with units).

3. **Electronic / Magnetic Ground State Extraction (`electronic_or_magnetic_phases`):**

   - Identify all explicit statements about electronic or magnetic phases for the main material AND related materials.
   - Recognized phases: band gap/semiconductor, Mott insulator, antiferromagnet, ferromagnet, charge density wave, spin glass, metallic state, paramagnetic/diamagnetic, etc.

```
              - Extract:  material, phase name, temperature (if stated), and type
                (transition vs measurement).
      4. **Descriptive Note ('missing_or_notable_information'):**
              - Provide a short (1{3 sentence) note summarizing important info
                that doesn't fit the fields, ambiguities, or extra details.
              - Must be grounded in search results.
      5. **General Constraints:**
              - Use ONLY information explicitly present in the search results.
              - If a Tc, phase, or temperature is not stated, return null (or
                empty list/string as appropriate).
              - Do NOT infer superconductivity or phases even if they are typical
                for the material class.
              - **Source DOIs**:  For each source, provide an object with:
                'title', 'authors' (list), 'year' (int), 'doi', and 'quoted_span'
                (relevant text snippet supporting the extraction).
              - Provide ONLY the JSON. No Markdown fencing or explanation.
```

# G    ADDITIONAL RESULTS

## G.1    MULTI-PROPERTY EXTRACTION

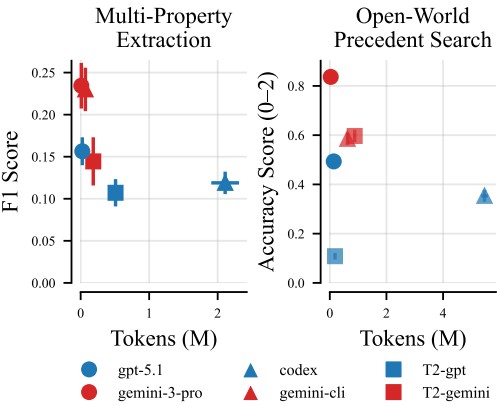

Figure 10: **Accuracy versus total tokens.** For SuperCon and Biosurfactants, we report mean ± 1 standard error recovery rate across 17 and 27 tasks, respectively. For Flux and Tc, we report the mean ± 1 standard error accuracy score across 50 and 200 tasks, respectively. Lighter points represent GPT-5-mini and Gemini-3-Flash.

Table 6: **F1 scores under varying numerical tolerances for property extraction.** We report F1 scores at 0.1% (strict), 1%, and 10% tolerance levels, alongside the property-material (PM) match which ignores value correctness entirely. Even with 10% tolerance, F1 scores remain substantially below PM match, confirming that precise value extraction is the primary bottleneck. Mean ± 1 SE.

| Agent | Model | F1 (0.1%) | F1 (1%) | F1 (10%) | PM Match |
|---|---|---|---|---|---|
| zeroshot | gemini-3-pro | $0.27 \pm 0.01$ | $0.28 \pm 0.01$ | $0.31 \pm 0.01$ | $0.43 \pm 0.02$ |
| gemini-cli | gemini-3-pro | $0.27 \pm 0.01$ | $0.28 \pm 0.01$ | $0.32 \pm 0.01$ | $0.43 \pm 0.02$ |
| T2 | gemini-3-pro | $0.14 \pm 0.03$ | $0.15 \pm 0.03$ | $0.16 \pm 0.03$ | $0.22 \pm 0.04$ |
| zeroshot | gpt-5.2 | $0.18 \pm 0.01$ | $0.19 \pm 0.01$ | $0.22 \pm 0.01$ | $0.35 \pm 0.01$ |
| codex | gpt-5.2 | $0.13 \pm 0.01$ | $0.14 \pm 0.01$ | $0.16 \pm 0.01$ | $0.26 \pm 0.01$ |
| T2 | gpt-5.2 | $0.14 \pm 0.01$ | $0.15 \pm 0.01$ | $0.17 \pm 0.01$ | $0.26 \pm 0.01$ |

## G.2 Open-World Precedent Search

Table 7: **Performance metrics for precedent search (mean ± 1 standard error over all trials).** Failure rate is the percentage of trials with errors or malformed outputs; Coverage is percentage of valid (not failed) trials with definitive answers (excluding "Unknown" responses); Accuracy (score=2) is percentage of valid trials with perfect scores; Accuracy (score=1) is the percentage of valid trials with correct classification, wrong value. Citation matches are reported as the percentage of trials with a DOI/title/any match (details in D)

| Agent | Model | Task performance (%) | | | | Citation Match (%) | | |
|---|---|---|---|---|---|---|---|---|
| | | Failure Rate | Coverage | Acc. (score=2) | Acc. (score=1) | DOI | Title | Any |
| gemini-cli | Gemini-3-Pro | 3.50 ± 0.75 | 62.52 ± 2.01 | 6.91 ± 1.05 | 46.80 ± 2.08 | 9.83 ± 1.22 | 10.67 ± 1.26 | 13.33 ± 1.39 |
| codex | GPT-5.1 | 10.67 ± 1.26 | 38.62 ± 2.10 | 6.72 ± 1.08 | 26.12 ± 1.90 | 17.00 ± 1.53 | 17.67 ± 1.56 | 17.67 ± 1.56 |
| terminus-2 | Gemini-3-Pro | 3.33 ± 0.73 | 62.41 ± 2.01 | 7.41 ± 1.09 | 46.90 ± 2.07 | 12.33 ± 1.34 | 16.67 ± 1.52 | 17.00 ± 1.53 |
| terminus-2 | GPT-5.1 | 6.83 ± 1.03 | 11.63 ± 1.36 | 0.72 ± 0.36 | 10.20 ± 1.28 | 3.17 ± 0.72 | 3.33 ± 0.73 | 3.50 ± 0.75 |
| qwen-code | Qwen3-Max | 3.67 ± 0.77 | 34.08 ± 1.97 | 5.19 ± 0.92 | 20.76 ± 1.69 | 1.67 ± 0.52 | 6.83 ± 1.03 | 7.17 ± 1.05 |
| No Agent | Gemini-3-Pro | 2.33 ± 0.62 | 79.35 ± 1.67 | 15.70 ± 1.50 | 54.27 ± 2.06 | 21.00 ± 1.66 | 29.33 ± 1.86 | 30.17 ± 1.88 |
| No Agent | GPT-5.1 | 0.67 ± 0.33 | 44.97 ± 2.04 | 9.40 ± 1.20 | 30.87 ± 1.89 | 24.83 ± 1.77 | 26.67 ± 1.81 | 26.67 ± 1.81 |

Table 8: **Efficiency metrics for precedent search (mean ± 1 standard error).** We note that `qwen-code` does not provide run metadata such as tool calls and token usage like other agent harnesses.

| Agent | Model | Tool Calls | Time Taken (s) | Total Tokens (M) | Cost (USD) |
|---|---|---|---|---|---|
| gemini-cli | Gemini-3-Pro | 24.8 ± 0.4 | 448.3 ± 14.7 | 0.624 ± 0.014 | 0.87 ± 0.02 |
| codex | GPT-5.1 | 58.5 ± 0.9 | 642.1 ± 9.1 | 5.464 ± 0.125 | 4.07 ± 0.09 |
| terminus-2 | Gemini-3-Pro | 42.4 ± 0.8 | 519.1 ± 9.9 | 0.878 ± 0.027 | 0.52 ± 0.01 |
| terminus-2 | GPT-5.1 | 40.7 ± 1.1 | 698.6 ± 12.3 | 0.183 ± 0.005 | 0.32 ± 0.01 |
| qwen-code | Qwen3-Max | – | 166.9 ± 5.1 | – | – |
| No Agent | Gemini-3-Pro | 1.0 ± 0.0 | 146.9 ± 2.8 | 0.032 ± 0.000 | 0.04 ± 0.00 |
| No Agent | GPT-5.1 | 29.8 ± 0.3 | 564.2 ± 13.1 | 0.139 ± 0.000 | 0.38 ± 0.00 |

Table 9: **Precedent search precision.** Score distribution is on *answered trials only* (mean % ± 1 standard error). S2 indicates correct class, correct value; S1 indicates correct class, wrong value; S0: incorrect class.

| Agent | Model | S0 (%) | S1 (%) | S2 (%) |
|---|---|---|---|---|
| gemini-cli | Gemini-3-Pro | 14.09 ± 1.83 | 74.86 ± 2.28 | 11.05 ± 1.65 |
| codex | GPT-5.1 | 14.98 ± 2.49 | 67.63 ± 3.26 | 17.39 ± 2.64 |
| terminus-2 | Gemini-3-Pro | 12.98 ± 1.77 | 75.14 ± 2.27 | 11.88 ± 1.70 |
| terminus-2 | GPT-5.1 | 6.15 ± 3.00 | 87.69 ± 4.11 | 6.15 ± 3.00 |
| qwen-code | Qwen3-Max | 23.86 ± 3.04 | 60.91 ± 3.49 | 15.23 ± 2.57 |
| No Agent | Gemini-3-Pro | 11.83 ± 1.50 | 68.39 ± 2.16 | 19.78 ± 1.85 |
| No Agent | GPT-5.1 | 10.45 ± 1.87 | 68.66 ± 2.84 | 20.90 ± 2.49 |

Table 10: **Citation match reliability for** $T_c$ **(Mean ± 1 standard error).** *Citation Quality* reports among trials with any correct citation match (DOI or title), the percentage that fall into corresponding scores. *Attribution Quality* reports among trials in a given score, the percentage that achieved any correct citation match.

| Agent | Model | Citation Quality (%) | | | Attribution Quality (%) | | |
|---|---|---|---|---|---|---|---|
| | | Score 0 | Score 1 | Score 2 | Score 0 | Score 1 | Score 2 |
| gemini-cli | Gemini-3-Pro | 16.25 ± 4.15 | 63.75 ± 5.41 | 20.00 ± 4.50 | 4.50 ± 1.22 | 18.82 ± 2.38 | 40.00 ± 7.84 |
| codex | GPT-5.1 | 35.85 ± 4.68 | 41.51 ± 4.81 | 22.64 ± 4.08 | 8.96 ± 1.39 | 31.43 ± 3.94 | 66.67 ± 7.97 |
| terminus-2 | Gemini-3-Pro | 22.55 ± 4.16 | 54.90 ± 4.95 | 22.55 ± 4.16 | 8.07 ± 1.62 | 20.59 ± 2.46 | 53.49 ± 7.70 |
| terminus-2 | GPT-5.1 | 57.14 ± 11.07 | 38.10 ± 10.86 | 4.76 ± 4.76 | 2.23 ± 0.64 | 14.04 ± 4.64 | 25.00 ± 25.00 |
| qwen-code | Qwen3-Max | 23.26 ± 6.52 | 48.84 ± 7.71 | 27.91 ± 6.92 | 2.22 ± 0.70 | 17.50 ± 3.48 | 40.00 ± 9.10 |
| No Agent | Gemini-3-Pro | 14.36 ± 2.61 | 56.35 ± 3.70 | 29.28 ± 3.39 | 13.68 ± 2.50 | 32.08 ± 2.62 | 57.61 ± 5.18 |
| No Agent | GPT-5.1 | 39.38 ± 3.87 | 36.88 ± 3.83 | 23.75 ± 3.37 | 17.50 ± 2.01 | 32.07 ± 3.45 | 67.86 ± 6.30 |

