# OpenReview forum: "The Reliability Gap in Agentic Evidence Verification for Materials Science"
_ICLR.cc/2026/Workshop/FM4Science — ICLR 2026 Workshop FM4Science Poster_

### Official Review · Reviewer_KUsR · 2026-02-13
**Evaluating the "Reliability Gap": A Rigorous Benchmark for Agentic Evidence Verification in Materials Science**

**Rating:** 8
**Confidence:** 4

**Review:**

**Summary**
This paper introduces a specialized evaluation framework designed to measure the factual reliability of general-purpose LLM agents in the domain of materials science. Unlike traditional benchmarks that use "sanitized" or "oracle" data, this study focuses on two real-world, high-noise workflows: Multi-Property Extraction from complex, multimodal PDFs and Open-World Precedent Search on the live web. Grounded in the SuperCon database, the researchers demonstrate a significant "reliability gap," where agents can identify relevant materials but struggle to extract precise numerical values or provide valid source attribution.

**Pros**

*1. Originality and Realism:* Shifts evaluation from controlled "exam-style" environments to the messy reality of scientific research, including the critical ability to verify "negative results" (evidence of absence).

*2. Domain-Specific Rigor:* Develops evaluation criteria with materials science experts, implementing a strict 0.1% SI unit threshold for numerical accuracy—a level of precision required for real-world laboratory utility.

*3. Infrastructure:* Utilizes Harbor, a sandboxed environment that allows for detailed trajectory logging, enabling a granular analysis of how agents use tools and where they fail.

*4. Significant Findings:* Challenges the assumption that increased agentic complexity (reasoning loops/scaffolding) improves performance, finding that simpler workflows or direct prompting often yield better accuracy-cost tradeoffs.

**Cons**

*1. Limited Scope of Models:* While frontier models like Gemini 3 Pro and GPT-5.2 are tested, the benchmark is primarily diagnostic; it identifies the gap but does not propose a novel architectural solution to bridge it.

*2. Dataset Size:* The evaluation is grounded on 200 papers. While meticulously curated, scaling this to a larger or more diverse set of scientific disciplines would strengthen the generalizability of the "reliability gap" thesis.

*3. Cost and Efficiency:* The paper highlights that agentic scaffolding often increases cost without increasing accuracy, which might limit the immediate practical adoption of these systems in resource-constrained academic labs.

Overall, this work is highly significant for the ML community as it provides a challenging, high-value testbed rooted in actual scientific bottlenecks. The finding that "successful retrieval does not imply faithful extraction" is a crucial insight for those developing RAG (Retrieval-Augmented Generation) systems. The paper is clearly written, with a robust methodology for matching property names using embedding models and LLM-as-judge prompts.

---

### Official Review · Reviewer_nrc2 · 2026-02-14
**The Hard Evaluation of "Soft" Evidence: Assessing the Rigor of Agentic Workflows on a Raw Scientific PDF Corpus**

**Rating:** 6
**Confidence:** 4

**Review:**

### **1. Summary**
This work provides a valuable and thorough assessment of LLM agents within the domain of materials science, specifically through the lenses of multi-property extraction and open-world search. The study effectively highlights the performance bottlenecks of current multi-step reasoning frameworks when faced with high-precision data. However, from a technical standpoint, the paper presents more as an extensive benchmarking exercise or 'stress test' rather than a novel algorithmic contribution. While the identification of this 'reliability gap' is significant, the manuscript would be substantially strengthened by the introduction of a novel architectural component or a theoretical model designed to mitigate these identified challenges.

### **2. Quality, Clarity, Originality, and Significance**
**Quality:**
The experimental execution is technically sound and thorough. However, the empirical findings are largely coupled with the specific characteristics of the SuperCon dataset, which may limit the generalizability of the results across other scientific domains. Furthermore, while the inclusion of "negative results" is highly commendable and relevant, the discussion remains primarily at the observational level.

**Clarity:**
The manuscript is well-organized and follows a logical progression. The objectives, task descriptions, and results are presented in a clear and accessible manner, allowing the reader to follow the experimental pipeline without difficulty.

**Originality:**
The study addresses critical real-world pain points, such as PDF parsing fragility and numerical hallucinations in scientific AI. However, these challenges are well-documented within the broader LLM and Agentic research communities. While applying these problems to the materials science pipeline is useful, the technical novelty of the proposed framework feels somewhat incremental.

**Significance:**
The work holds clear value for the materials informatics community, providing a much-needed reality check on the current state of autonomous evidence verification.


### **3. Pros:**

- Valuable Dataset Contribution: The curation of a dataset comprising 200 raw, expert-annotated scientific PDFs from the SuperCon database is a commendable effort. This resource provides a practical foundation for future research in the domain of materials informatics and evidence verification.

- Practical and Realistic Perspective: By shifting away from "idealized" or "sanitized" data environments and directly addressing the complexities of raw PDF parsing, the study offers a pragmatic evaluation that closely mirrors the real-world challenges faced by human researchers.

### **4. Cons:**

- Opportunities for Algorithmic Innovation: While the study successfully implements an "Open-World Precedent Search" using existing agentic frameworks, the technical contribution remains primarily at the application and characterization level. The manuscript would be further elevated by unique architectural modifications designed specifically to bridge the identified reliability gaps.

---

### Meta-Review · Area_Chair_7jXE · 2026-02-27

**Recommendation:** Accept (Poster)
**Confidence:** 4

**Metareview:**

The average review score is above 6, which means reviewers recommended an acceptance.

---

### Decision · Program_Chairs · 2026-03-03

Accept (Poster)